# Amplitude-Resolved Single Particle Spectrophotometry: A Robust Tool for High-Throughput Size Characterization of Plasmonic Nanoparticles

**DOI:** 10.3390/nano13172401

**Published:** 2023-08-23

**Authors:** Rodrigo Calvo, Valerio Pini, Andreas Thon, Asis Saad, Antonio Salvador-Matar, Miguel Manso Silván, Óscar Ahumada

**Affiliations:** 1Mecwins S.A., Ronda de Poniente, 15 2°D, Tres Cantos, 28760 Madrid, Spain; 2Departamento de Física Aplicada, Universidad Autónoma de Madrid, Campus de Cantoblanco, 28049 Madrid, Spain; 3Centro de Microanálisis de Materiales, Universidad Autónoma de Madrid, Campus de Cantoblanco, 28049 Madrid, Spain

**Keywords:** plasmonic, nanoparticles, metrology

## Abstract

Plasmonic nanoparticles have a wide range of applications in science and industry. Despite the numerous synthesis methods reported in the literature over the last decades, achieving precise control over the size and shape of large nanoparticle populations remains a challenge. Since variations in size and shape significantly affect the plasmonic properties of nanoparticles, accurate metrological techniques to characterize their morphological features are essential. Here, we present a novel spectrophotometric method, called Amplitude-Resolved Single Particle Spectrophotometry, that can measure the individual sizes of thousands of particles with nanometric accuracy in just a few minutes. This new method, based on the measurement of the scattering amplitude of each nanoparticle, overcomes some of the limitations observed in previous works and theoretically allows the characterization of nanoparticles of any size with a simple extra calibration step. As proof of concept, we characterized thousands of spherical nanoparticles of different sizes. This new method shows excellent accuracy, with less than a 3% discrepancy in direct comparison with transmission electron microscopy. Although the effectiveness of this method has been demonstrated with spherical nanoparticles, its real strength lies in its adaptability to more complex geometries by using an alternative analytical method to the one described here.

## 1. Introduction

Plasmonic particles are nanometric structures of metallic materials or metamaterials that possess exceptional optical properties [1,2] and have attracted considerable attention in recent years in many areas of biomedicine, including biosensing [3] and drug delivery [4,5], as well as in other fields such as catalysis [6] and electronics [7]. When radiation strikes these structures, it induces an oscillation of the free electrons, known as localized surface plasmon resonance (LSPR), which leads to a remarkable scattering signal at certain wavelengths. What makes these types of structures interesting in various fields is the ability to tune their spectral responses by introducing small geometrical and optical variations, such as the refractive index of the medium [8], the particle shapes [9], their compositions [10], and, most critically, their sizes [11,12]. Given the high sensitivity of these structures to any morphological changes, a precise control of the fabrication process is required to achieve a good spectral homogeneity.

Several methods, including both bottom-up [13] and top-down [14,15] approaches, can be used to fabricate plasmonic nanoparticles and are being continuously improved to produce nanoparticle populations with very narrow size distributions. However, achieving greater precision in nanoparticle size typically requires more costly synthesis processes, and the coefficient of the size variation can vary by up to 50% between batches [16]. Consequently, metrology systems have become an essential tool for analysis and quality control in routine nanoparticle production.

Scanning electron microscopy (SEM) and transmission electron microscopy (TEM) are widely used metrology systems in laboratories due to their nanometric resolution, which allows the measurement of structures smaller than 1 nm [17]. However, it is important to note that these techniques often present additional challenges due to their complexity of operation and maintenance. Electron microscopes require a constant vacuum to operate and a continuous high voltage source during measurements. These complex and expensive instruments often require trained personnel to operate them. As a result, these techniques are costly, complicated, and time-consuming compared to optical microscopy methods.

Several techniques based on optical spectrophotometry offer a cheaper and simpler alternative to electron microscopy for determining the size of plasmonic nanoparticles.

Dynamic Light Scattering (DLS) is a time-resolved optical method that uses laser illumination to measure the size of colloidal nanoparticles by the absorption peak of light scattered by all particles present in a batch [18,19]. This is a rapid characterization method where sample preparation is simple as long as the particles are suspended in a colloidal state. However, it is important to note that derived measurements cannot provide the exact size distribution of a batch, because measurements are performed on the entire colloidal solution, as the main parameter obtained from DLS is the hydrodynamic radius. The hydrodynamic radius refers to the ‘effective’ size of the nanoparticle or as it moves through a fluid. This size includes not only the actual size of the particle but also the layer of solvent that moves with the particle as it diffuses through the fluid. Therefore, the final spectrum obtained is averaged over the entire nanoparticle population. If the batch has a broad or very narrow size distribution, this method can only provide qualitative information. DLS is able to distinguish the types of particles present in the colloidal sample with a poor resolution, so any contamination or aggregation of particles can lead to inaccurate size measurements. In addition, the viscosity and refractive index of the solvent can also affect the size of a given batch. The size distribution results obtained by DLS are averaged measurement of the whole colloidal sample and do not provide information about the physical properties of each individual nanoparticle [20].

Another valuable alternative is scanning flow cytometry (SFC) [21]. SFC is a time-resolved technique capable of measuring the full angular dependence of the light-scattering intensity, using a combined optical and hydrodynamic system to measure the fluorescence and scattering signal from individual particles. This approach provides detailed insight into the size, shape, and composition of a batch of nanoparticles. Unlike DLS, this technique can measure individual particles with high throughput, meaning that the information obtained is a true size distribution.

However, the experimental setup is inherently more complex, requiring a monochromatic laser that provides less spectral information but also a hydrodynamic framework that requires careful control of the variables associated with the sample liquid. This limitation restricts the method to the measurement of large nanoparticles, between 0.9 and 15 µm in size.

We have recently introduced a novel measurement technique based on dark-field microspectrophotometry that allows the size of thousands of individual nanoparticles to be assessed in a matter of minutes [22,23]. This method allows the size of individual nanoparticles to be determined indirectly by measuring the wavelength of the plasmon resonance peak [24,25]. Compared to TEM, the new method presented here shows a discrepancy in the average nanoparticle size measurement of less than 3%. Here, we refer to ‘discrepancy’ as the percentage variation between the absolute mean values obtained by both techniques, observed over all the batches discussed in the manuscript. Despite its speed and sensitivity, the main limitation of this technique is fundamental in nature, as the spectral peak shifts for small particles below 50 nm are extremely small, posing a challenge for the accurate measurement of smaller nanoparticles.

In this study, we demonstrate that the above limitations can be overcome by introducing a novel metrological method called Amplitude-Resolved Single Particle Spectrophotometry (AR-SPS). This technique is a straightforward, high-throughput approach capable of quantifying particles smaller than 50 nm in just a few minutes. Unlike DLS, this technique provides detailed information about the size distribution of thousands of individually measured nanoparticles.

## 2. Materials and Methods

Experimental Setup. The practical realization of the microspectrophotometer was achieved by using a 100 W halogen lamp and a VI-IR electrooptical filter (Thorlabs, Mölndal, Sweden, Kurios WB1, with a working spectral range of 400 nm to 700 nm), coupled with a mechanical adapter to the diascopic arm and dark-field condenser of a commercial optical microscope (Nikon Eclipse Ni-U) using a 20× dark-field objective (NA 0.4, MUE 61200 from Nikon, Minato City, Japan). A scientific-grade monochromatic CMOS camera (Ximea MC050MG-SY) was used to acquire the measurements. In addition, microscopic measurements were obtained using an episcopic illuminator coupled with a 50 W halogen lamp, which was captured by a RGB CMOS camera (Ximea, Münster, Germany, MD120CU-SY). The hardware components of the instrument, including the RGB and monochromatic cameras, the tunable optical filter, and the halogen lamps, were all controlled by proprietary software, developed with LabVIEW 2020.

Sample preparation. Spherical gold nanoparticles from Nanopartz (Salt Lake City, UT, USA) were used in the experiments, with nine different nominal diameters ranging from 40 to 150 nm. Each batch of nanoparticles was initially dispersed in milliQ^®^ water at a concentration in the mg/mL range. To ensure good nanoparticle monodispersion and sufficient statistics with a single image acquisition, the nanoparticles were resuspended in milliQ^®^ water at a concentration of 200 µg/µL. To improve nanoparticle monodispersity, the colloidal solution was sonicated for 5 min and vortexed for 5 min prior to sample preparation. A 50 µL drop of the solution was then applied to a transparent glass slide (Thermo Fisher, Waltham, MA, USA, microscope slides, Menzel Gläser, Braunschweig, Germany) and allowed to dry at room temperature. Before the glass coverslip (Menzel Deckgläser 20 × 20 mm, 170 µm thick) was applied to the sample substrate, a drop of 2 µL of glycerol (Sigma Aldrich, St. Louis, MI, USA) was added to the dried GNPs. This addition of glycerol reduced the optical mismatch between the refractive index of the glass substrate and the surrounding environment.

## 3. Results and Discussion

For sizes smaller than the illuminating wavelength, the scattering amplitude scales approximately as *~d*^6^ [26]. It is an optical property that scales strongly at any nanoparticle size level, allowing the accurate measurement of nanoparticle sizes even smaller than 50 nm. This approximation is valid for non-agglomerated individual nanoparticles that are not in a polydisperse state [27]. The only limitation of this novel approach is the instrumental capability of the equipment used to detect and measure nanoparticles, i.e., it is necessary to improve the signal-to-background ratio as much as possible. As the nanoparticles analyzed here are beyond the Rayleigh regime (i.e., sizes above ~10 nm), they exhibit anisotropic emission, with the forward scattering signal being significantly larger than the backward scattering contribution [28,29]. For this reason, the use of a transmission detection system is more advantageous than the episcopic system used in the previous work.

A schematic drawing of the experimental setup is shown in Figure 1. The white light coming from a halogen lamp is directed at an electrooptical filter, which separates the light into its constituent wavelengths. The filtered light is directed to a dark-field condenser that focuses it onto the surface of the transparent sample where the nanoparticles are located. The scattered light is collected by a dark-field objective and redirected to a monochromatic CMOS camera placed at the image plane of the experimental setup, which collects the images at each wavelength of illumination (see Materials and Methods for more technical details). The experimental setup is also equipped with an episcopic illuminator capable of imaging the same sample region.

The measurement of the scattering amplitude is more complex than the spectral measurement of the wavelength, because it depends not only on the morphological and optical characteristics of the particle but also on the measurement system used to obtain the signal (such as the numerical aperture of the objective [30], the angle of illumination, [31] the polarization of the light, etc.). Therefore, it is essential to use a batch of nanoparticles measured under the same conditions as a calibration system in order to eliminate the influence of any instrumental dependencies coming from the AR-SPS setup.

Each plasmonic nanostructure has a specific relationship between its scattering amplitude and its size, which can be determined by either analytical or semi-numerical methods [32]; for example, in the case of spherical geometries, a well-defined relationship between the size *d* of spherical nanoparticles and their scattering amplitude *A* can be established thanks to the Mie theory [25].
(1)d≈b· ln1−AAref.αdref.βc
where *α*, *β*, *b*, and *c* are constants depending on the optical properties of the medium and the nanoparticles themselves (for more information on the theoretical derivation (see the Appendix A) [24,33,34], and *d_ref_* is the diameter of the reference batch. The scattering amplitude *A* in Equation (1) has been normalized by *A_ref_*, which corresponds to the scattering amplitude of the nanoparticles of the reference lot. Normalization is a key aspect of this method, as it removes the influence of measurement system dependencies.

It should be emphasized that the formula obtained in Equation (1), as already established, was finally obtained from a numerical derivation after introducing the geometrical and optical properties of the nanoparticles and the surrounding media. The same theoretical framework is still valid for spherical nanoparticles with different plasmonic materials and the surrounding media (see the details in the Appendix A). In the case of nanoparticles with geometries more complex than a sphere, this approach is still valid but by using an extended theoretical framework [35,36].

The AR-SPS includes a simple and rapid sample preparation based on the drop casting method [22,37]. A schematic of the sample preparation is shown in Figure 2a; a drop of 50 µL was placed on a microscope slide and dried at room temperature. A drop of 2 µL of glycerol 28 was added to the dried GNPs before the glass coverslip was placed on the sample substrate. The addition of glycerol reduces the optical mismatch between the refractive index of the glass substrate and that of the surrounding environment, thus avoiding the splitting of substrate-mediated plasmonic modes and allowing the optical response to be well predicted by the standard Mie theory.

Despite the careful sample preparation described in sample preparation section, drop-casting drying is a complex process that depends not only on the properties of the liquid and the particles suspended on it but also on the properties of the surface [38]. Consequently, even with the most precisely controlled procedures, particle agglomerations (such as dimers, trimers, etc.) can occur, making it difficult to accurately size individual nanoparticles. Because the plasmonic response of the aggregates is very different from that of the individual particles, due to the interparticle plasmonic interaction, they can be easily distinguished simply by their color and brightness, without the need for a full spectrum, which can be prefiltered by microscopy measurements. Before proceeding with measurements, it is necessary to take a RGB microscopy image with the color camera, as described in the experimental setup, to distinguish monomers from other particles. By using a custom algorithm that discriminates the brightness and color of each individual optical spot on the surface, it is possible to identify monomers from other particles (see the Appendix A for further explanations of monomer discrimination).

The ability to distinguish individual particles from agglomerates and contaminants is a major advantage over techniques such as DLS, where the presence of such formations cannot be distinguished from the particles being measured, resulting in a size measurement that can be inaccurate.

Once the sample is prepared, dark-field microspectrophotometry is performed by sequentially illuminating the sample at different wavelengths, as shown in Figure 2b. Cropped images of the spectral measurements of 100 nm GNPs are shown, while the normalized spectra of approximately 1000 nanoparticles are shown in Figure 2c. For the sake of clarity, normalized scattering spectra have always been obtained in this paper by dividing the scattering signal of each nanoparticle by the scattering signal coming from the substrate; this type of normalization allows obtaining a direct estimate of the signal-to-noise ratio for each nanoparticle measured (for more technical information on data processing, see the Appendix A).

In good agreement with the Mie theory, we can observe how the resonance peak reaches its maximum at about 575 nm. Although the size variability of the studied nanoparticles is about 4%, the brightness variability is as high as 22%. This increased relative uncertainty in the amplitude is expected, because, as mentioned above, the amplitude scales with the diameter by about a power of six (i.e., *∂A⁄A ≈ 6 ∂d⁄d*) [26]. Therefore, the scattering amplitude of each individual nanoparticle can be calculated by performing a Lorentzian fit to each individual particle spectrum (for more details on data analysis, see the Appendix A).

We have demonstrated the effectiveness of the AR-SPS technique in a proof-of-concept experiment using spherical nanoparticles with diameters ranging from 40 to 150 nm (Nanopartz, Inc.). In total, we used nine different batches of nanoparticles for these experiments. Although the experiments were performed with spherical nanoparticles, it is important to note that it is possible to also extend this method to other shapes, such as nanorods [39], shells [40], and other geometries [41], as long as a relationship between the particle size and its plasmonic emission can be calculated, either analytically or numerically [32]. Therefore, the applicability of our technique extends beyond the realm of spherical particles and introduces the potential for further exploration in the study of other nanoparticle geometries. For example, in the case of nanorods, due to their shape, they have a transverse and a longitudinal emission mode, which results in a scattered emission spectrum consisting of two peaks with different amplitudes that can be used to characterize their aspect ratio.

Experimental data on how AR-SPS can be used to characterize the aspect ratio of individual gold nanorods can be found in the Appendix A.

Figure 3a shows the normalized scattering spectra of individual particles for each of the characterized batches. The data show that, while the plasmonic peak undergoes minimal spectral changes with the decreasing nanoparticle size, the amplitude signal consistently increases in magnitude for all measured sizes.

In Figure 3b, the experimental data of each batch were directly compared with the theoretical Equation (1), using the 90 nm batch as the reference sample. Since we considered spherical nanoparticles immersed in glycerol in the following experiments, the constants used in Equation (1) are as follows: α = 3.21 × 10^7^, β = −3.52, b = −37.85 nm, and c = 14.58. These numerical values are valid as long as we express *d_ref_* in nm in Equation (1) and change by using different combinations of media and plasmonic materials (see more details in the Appendix A).

To improve the comparison between experiment and theory, the TEM nanoparticle size was used instead of the nominal size in Figure 3b (see the Appendix A for more technical details). The amplitude data points shown here are experimental values coming from approximately 5000 individual nanoparticles spectra for each analyzed batch measured with the AR-SPS, with the error bars representing the corresponding standard deviation. Although the sizes of all the batches are characterized by TEM, for simplicity, the *d_ref_* used for the reference batch is the nominal value of 90 nm, avoiding any previous measurements by TEM. The black dashed line in Figure 3b represents the Mie theory scattering amplitude values for GNPs ranging from 30 to 140 nm, obtained by using Equation (1); the excellent agreement observed between the experimental data and the theoretical predictions is a strong indication of the effectiveness of the AR-SPS method.

The relationship between the diameter and scattering amplitude observed in Figure 3b shows that the main limitation of the method is not theoretical but due to the limitations of the experimental setup. Although we measured particles down to 40 nm in the following experiments, we could have gone even lower by improving the signal-to-noise ratio of the instrument; this could be achieved, for example, by using objectives with a larger numerical aperture or by amplifying the scattering signal using multi-dielectric substrates [42,43] or a detection scheme based on the total internal reflection [44,45].

To evaluate the accuracy of the proposed method in estimating nanoparticle diameters, a direct comparison with TEM is necessary. Figure 4 shows a correlation between the diameter obtained by TEM and the diameter estimated by AR-SPS (for more technical details, see the Appendix A). The error bars on both the x- and y-axes represent the standard deviations derived from the TEM and amplitude data, respectively. At least 500 particles per batch were characterized by TEM, and more than 5000 individual nanoparticle spectra were obtained for the amplitude data.

It is noteworthy that the experimental data closely followed the ideal correlation curve (black dashed line in Figure 4), indicating excellent agreement between the diameter estimated from the AR-SPS and the true diameter obtained from TEM, with an average discrepancy of about 2.6%.

Our initial experiments primarily utilized batches of nanoparticles characterized by a single size, with slight variations around this central size, but since the AR-SPS has the capability to measure the spectra of individual nanoparticles, it is also capable of multiplexing i.e., measuring samples with GNPs of different sizes, shapes, or materials. In the Appendix A, we added data from an experiment in which batches of six different sizes were mixed.

## 4. Conclusions

In summary, our study introduces a new metrological characterization method called AR-SPS that uses the amplitude of the scattering signal to estimate the nanoparticle size. This new method overcomes previously observed limitations, because the scattering amplitude is a spectral fingerprint that scales strongly with the size at each nanoparticle size level, allowing even the smallest nanoparticle sizes to be measured. The method’s simple sample preparation and use of commonly available dark-field microscopy techniques allow for rapid, high-throughput nanoparticle size estimations. We have demonstrated the feasibility of our method with a proof-of-concept experiment consisting of the characterization of nine different batches of spherical nanoparticles ranging in diameter from 40 nm to 150 nm. Having shown that the experimental data are in excellent agreement with the theoretical predictions of the Mie scattering theory, we then demonstrated by direct comparison with TEM that this method achieves a size estimate with an uncertainty of less than 3%. AR-SPS also supposes an advantage over other optical size characterization methods, such as DLS, because it can provide accurate information about the size distribution of a sample. Although the feasibility of the method has been demonstrated here with spherical nanoparticles, the real strength of this technique is that it can be adapted to nanoparticles with more complex geometries, even core–shell geometries. Therefore, this method has broad potential in various fields, such as materials science, biotechnology, and nanomedicine, where accurate and rapid nanoparticle size characterization is critical.

The AR-SPS method serves as a viable alternative to existing nanoparticle size characterization methods, offering distinct advantages in certain scenarios. By adding value to specific applications, it represents a significant refinement that stands to improve both the efficiency and accuracy.

## Figures and Tables

**Figure 1 nanomaterials-13-02401-f001:**
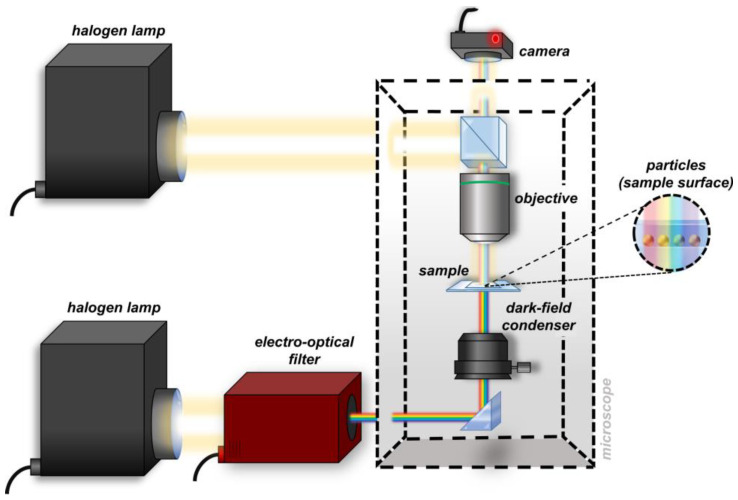
Schematic drawing of the AR-SPS experimental setup; the sample imaging is performed in reflection mode, while the sample spectrophotometry is performed in transmission mode.

**Figure 2 nanomaterials-13-02401-f002:**
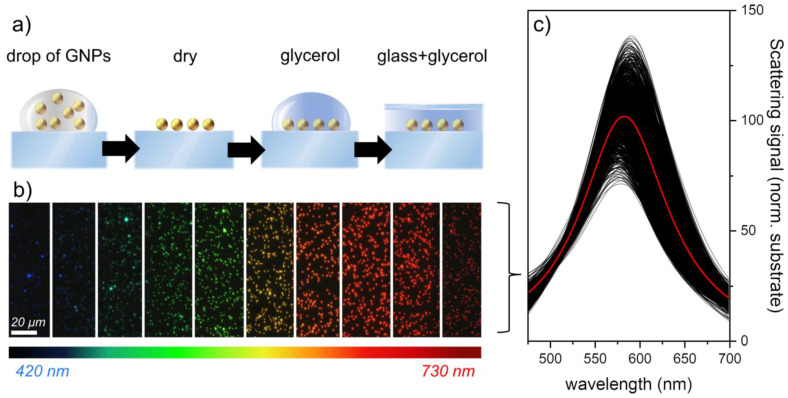
Sample preparation scheme, spectral images, and scattering spectra of gold nanoparticles (GNPs). (**a**) Schematic representation of the sample preparation process, in which a 50 μL drop of GNPs at a concentration of 200 μg/mL was deposited on a glass slide, followed by the addition of a 2 μL drop of glycerol and a coverslip. (**b**) Cropped spectral images obtained for 100 nm particles used to measure the scattering amplitude at different wavelengths. (**c**) Scattering spectra of approximately 1000 GNPs extracted from the spectral images shown in (**b**) and normalized to the sample background. The red line represents the average scattering amplitude at each wavelength. The average size of the GNPs, extracted from the amplitude using Equation (1), is 106 ± 10 nm.

**Figure 3 nanomaterials-13-02401-f003:**
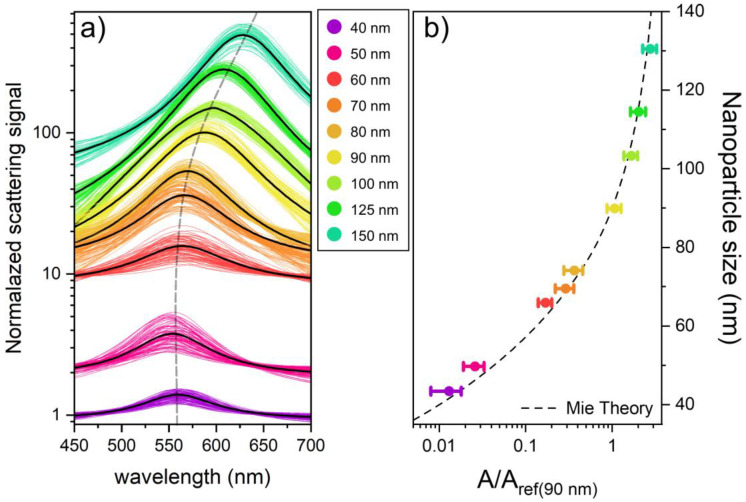
Spectral analysis and theoretical dependence of the scattering amplitude on the nanoparticle size. (**a**) Thousands of individual normalized particle spectra are obtained for each batch using the AR-SPS technique, with the mean value represented by the black line. The dotted line marks the spectral shift between batches, and all data have been normalized to the sample background. (**b**) This graph shows the normalized mean amplitude values for different batches of nanoparticles measured using AR-SPS, with each point value corresponding to a different nanoparticle size. Every point is normalized to a reference batch of 90 nm. The associated error bars show the standard deviation for each measurement. The nanoparticle sizes shown represent the mean values obtained by TEM characterization. A black dashed line on the graph illustrates the theoretical amplitude-to-reference ratio (*A/Aref*) as a function of the nanoparticle size, as derived from Equation (1).

**Figure 4 nanomaterials-13-02401-f004:**
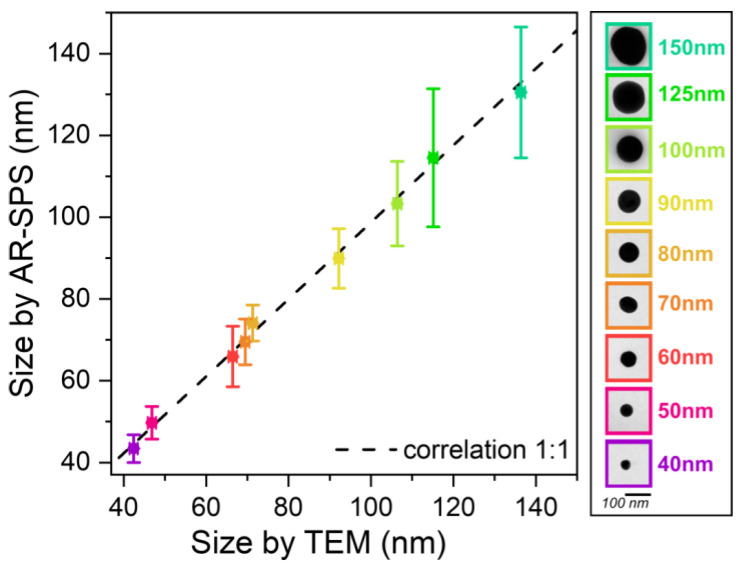
Correlation between nanoparticle size values obtained by AR-SPS and those obtained by TEM. The legend shows the nominal values for each batch, along with an image of a single nanoparticle taken by TEM. The points follow a very near-ideal linear relationship, represented by the black dashed line; in fact, when a linear fit is performed, the resulting slope has a value of 0.94.

## Data Availability

All details regarding theoretical derivations, data processing and analysis, TEM measurement and characterization procedures, and size comparisons between TEM and AR-SPS are provided in the text and Appendix A. Any clarifications can be obtained by contacting the corresponding author.

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
