# Peer review of "Amplitude-Resolved Single Particle Spectrophotometry: A Robust Tool for High-Throughput Size Characterization of Plasmonic Nanoparticles"

_nanomaterials, 2023, doi:10.3390/nano13172401_

Round 1

Reviewer 1 Report

Review of the manuscript “Amplitude-Resolved Single Particle Spectrophotometry: A Robust Tool for High-Throughput Size Characterization of Plasmonic Nanoparticles” by Rodrigo Calvo et al.

 The manuscript reports an Amplitude-Resolved Single Particle Spectrophotometry method to characterize the average particles size by analyzing the wavelength-dependent scattering patterns from thousands of individual particles dried on a glass substrate from a drop of sampling colloids. This manuscript may be of interest for Nanomaterials readers and could be published after major revisions indicated below.

1.      Abstract: “Although the effectiveness of this method has been demonstrated with spherical nanoparticles, its real strength lies in its adaptability to nanoparticles of arbitrary shape and geometry, making it an advantageous alternative to the gold standard of electron microscopy”.

I strongly disagree with this statement. The scattering by nonspherical particles and its geometrical characterization present significant problems. Specifically, the scattered amplitude depends not only on the particle shape but on its orientation as well. Without any preliminary information about the particle shape the AR-SPS method cannot be applied. Just consider such popular particles as Au nanostars and suggest any reasonable approaches to characterize them by AR-SPS. I suggest exclude this fragment from the Abstract. The same for lines 179-182.

2.      Clearly, the AR-SPS is not an absolute method and needs preliminary calibration, in contrast to absolute TEM and SEM methods (lines 123-125). This fact should be pointed out already in the Abstract.

3.      In the Introduction section (line 55) the authors mentioned “….complex sample preparation” of TEM and SEM. In my opinion, the sentence is at least not accurate. Indeed, all we need in our own sample preparation TEM practice is to place a drop of diluted colloids onto commercial TEM gird. That’s all. Please, compare to your sample preparation steps (line 82 and below> Figure 2). The sentence should be deleted.

4.      Lines 60-61: “Compared to standard electron microscopy, it shows an average discrepancy in nanoparticle size of less than 3%”. The sentence is not clear. Which discrepancy the authors have in mind: between TEM and AR-SPS, the STDs of TEM and AR-SPS. Please, revise the fragment.

5.      Some proper citations would be desirable in Introduction. First, the authors did not mention DLS method. Meanwhile, Malvern Zeta Nano sizers are common cheap and available instruments in many Labs over the world. Further more, DLS provides the particle size distributions (of course, they should be used with caution) in contrast to limited average-size characterization by AR-SPS. This point should be discussed in the revision. Second, the scanning flow citometry can be noted as an example of high-performance method [V.P.Maltsev Scanning flow cytometry for individual particle analysis Rev. Sci. Instruments 71, 243-255 (2000)]. Finally, a recent review “Synthesis and plasmonic tuning of gold and gold-silver nanoparticles”, Russ. Chem. Rev., 2022, 91 (10) RCR5058 could be cited in context of line 39-41.

6.      Sample preparation (lines 82-94). Most assailable step is drying of drop sample on a glass substrate. This process typically leads to the formation of clustered patterns and this step cannot be controlled properly. As a result, the dublets, triplets, etc. of particles can be formed thus leading to strongly enhanced scattering intensities. A common practice is to use SEM-Optics collocation techniques (for example, in SERS) to examine the source of recorded signals. The authors should provide convincing evidence for truly single-particle scattering in their study.

7.      Line 97: “Since the scattering amplitude scales with size [21] as ~d^6….”, The sentence is not accurate. For polydisperce system the size exponent is typically less than 6 for the average size less than 100 nm, and for d>100 nm it drastically decreases (see, e.g. Colloid J. 2003. V. 65, No. 5, p. 652-655. Doi: 1061-933X/03/6505).

8.      Lines 207-209. The agreement between dashed line and points is quite expected as the same eq.(1) has been used for calculation. Some comments are needed here.

9.      The low limit of AR-SPS is estimated to be 40 nm. For nanobiotechnology, the size range 10-40 nm is most important. Thus, the proposed method is not applicable here. The authors should emphasize the point.

10.  Minor revisions are needed in English. E.g. line 40 “the particles shape”. Please revise as the particlE shape or the shape of particlES.

Minor editing of English language required

Reviewer 2 Report

The manuscript presents an experimental method for determining the sizes of gold spherical nanoparticles in experiments with the scattering of monochromatic radiation in the visible range upon excitation of plasmon resonance; in this case, the plasmon resonant wavelength is red-shifted with an increase in the size of the nanoparticle. The sample is irradiated with filtered (monochromatic) radiation in the dark field experimental protocol, which allows authors to find the plasmon resonance wavelengths. To determine the size of gold nanoparticles, the authors need to find the scattering intensity Aref at the spectral maximum for some reference size dref of a gold nanoparticle. The results of measuring the sizes of nanoparticles obtained by this technique are confirmed by measurements with tunneling electron microscopy. Since the authors use gold nanoparticles of a known size, it is quite natural that a good correlation between the measurement results obtained by these two techniques is achieved. It should be noted that the theoretical substantiation of the methodology proposed by the authors is practically absent in the main text of the manuscript. Namely, formula (1), on the basis of which the authors estimate the particle sizes, is given in the main text. However, the derivation of this formula is presented in supplementary files. In supplementary files, the first three formulas are well known, whereas Eqn. (4), which relates the amplitude of the scattered signal and the size of a scatterer and which, according to the authors, can be deduced from formula (1) in supplementary files, is given without derivation.

I have certain doubts that this paper contains new fundamental results. In the case of metal nanoparticles, it is well known that only one resonant peak (plasmon resonance due to electro-dipolar scattering) appears in the scattering spectrum, and this resonance wavelength depends on the size of the scattering nanoparticles. Of course, it is possible to calibrate the measuring setup by gold nanoparticles of a known size dref, and then, using formula (1) and the calculated parameters α, ?, b and c, determine the sizes of gold nanoparticles under study. However, for nanoparticles made of another metal, it is necessary to find its own specific set of parameters α, ?, b and c. Thus, the proposed technique is not universal. Note that in experiments, as a rule, it is necessary to determine the sizes of not only metal nanoparticles with plasmon resonance, but also dielectric and semiconductor nanoparticles, in which there are no free electrons and for which, as a rule, several peaks in the Mie scattering spectrum, arisen due to electric and magnetic dipoles and quadrupoles, are observed. It is clear that the technique, suggested by the authors, is in principle unsuitable for such nanoparticles. However, to determine the size of nanoparticles in scattering experiments, there exists a traditional method of dynamic light scattering. This technique is used to determine the sizes of plasmonic, dielectric, semiconductor particles, as well as gas nanobubbles suspended in a liquid. Summing up, the worth of the methodology proposed by the authors is not quite clear to me. In my opinion, the dynamic light scattering technique is much better suited for the determination of nanoparticles suspended in a liquid. I would recommend the authors to rewrite the text of the manuscript, focusing on the description of the advantages of the proposed technique in comparison with the dynamic light scattering. The manuscript in its present form should not be published.

Round 2

Reviewer 1 Report

I am satisfied with the revised manuscript. It can be published as is.

Reviewer 2 Report

I have carefully read the response of the authors to my criticisms. In my first review I suggested to perform a comparative analysis of their methodology and the traditional Dynamic light scattering (DLS) technique. In what follows quotations from the authors' response are in italics. The authors write: This manuscript introduces a novel method that uses the measurement of the scattering amplitude to indirectly estimate the size of the nanoparticle. I agree that this is a new method, but I keep my previous point of view that this method is significantly worse than the traditional DLS method. The advantages of the proposed technique over DLS, noted by the authors in the new version of the manuscript, look absolutely far-fetched. The authors refer to works [18, 19] on DLS, which present the results of measurements of gold nanoparticles.

Remind that the DLS technique is based on measuring the correlation function of scattered light intensity. The scattering nanoparticles perform Brownian motion, and the correlation time is determined by the diffusion coefficient of nanoparticles in a liquid, that is, this time is determined by the viscosity of the liquid, temperature, and particle size. For particles of different sizes, distribution functions of sizes can be easily found. Trying to find some advantages of the proposed methodology compared to DLS, the authors write: However, it is important to note that measurements derived from DLS cannot provide the exact size distribution of a batch because measurements are performed on the entire colloidal solution. This statement is completely wrong! I would say that the method proposed by the authors, cannot determine the size of metal nanoparticles if this suspension is a poly-dispersed one. Further, the authors write: DLS is also unable to distinguish the types of particles present in the colloidal sample, so any contamination or aggregation of particles can lead to inaccurate size measurements. I agree with the statement that the type of nanoparticles in the DLS technique remains undefined. However, in real situations the researchers deal with poly-disperse suspensions of nanoparticles made of various metals. Thus, the researchers should know what kind of metal it is. Table S1 shows the parameters b, c, a and b for silver and gold nanoparticles. If I need to study a mixture of silver and gold nanoparticles with unknown concentrations, can I use the method proposed by the authors? The authors' next argument: In addition, the viscosity and refractive index of the solvent can also affect the size of a given batch. This results in an absolute measure of average emission that does not provide quantitative information on the exact size distribution [20]. However, I do not see any problem here, since all DLS devices contain the databases on the viscosity and refractive index of various liquids for a fixed temperature.

Summing up, the technique proposed by the authors is certainly less suitable for solving actual problems that arise in the study of colloidal systems, as compared with the DLS technique. As was mentioned, the DLS technique is based on the scattering of monochromatic laser radiation (typically the wavelength of 633 nm is used, i.e. there are no absorption and luminescence excitation effects for most liquids), and the scattering particles can have an arbitrary physical nature and various sizes. At the same time, in the method proposed by the authors, instead of a monochromatic laser source, it is necessary to use a broadband halogen lamp and an electro-optical filter (the authors do not explain what it is), that is, the optical system seems to be more complex. The experimental result in the DLS experiment is a series of particle size distributions. This, in particular, is also related to aggregates of nanoparticles. In principle, DLS devices can be used to measure the zeta potential of scatterers, that is, it is possible to determine the charge of nanoparticles, which is very important, for example, in biology. Note that in the papers [18, 19] cited by the authors, poly-disperse suspensions of gold nanoparticles were studied. For example, in [18], the aggregation of nanoparticles was investigated by the DLS technique. In [19], mixtures of 20-nm and 100-nm gold particles at various ratios were studied, see Fig. 7(a) of this work. At the same time, it is impossible to study poly-disperse suspensions of plasmonic nanoparticles using the technique proposed by the authors. As I noted in my first review of this manuscript, the main drawback of the authors' technique is the lack of universality. In the method proposed by the authors, it is impossible to determine the size of metal nanoparticles if the suspension of nanoparticles is not a monodispersed one. It seems to me that the results of works [18, 19] cited by the authors are much more interesting in the context of the applications for studying gold nanoparticles. I would like to quote here from [18]: “Compared to the surface plasmon resonance technique, DLS is a low cost and low maintenance instrument”. Summarizing, this manuscript should not be published.
